# Association between benign prostatic hyperplasia and suicide in South Korea: A nationwide retrospective cohort study

**Sang-Uk Lee[1], Sang Hyub Lee[2], Ah-Hyun So[1], Jong-Ik Park[3], Soojung Lee[4], In-Hwan Oh[5], Chang-Mo Oh**[5]*

1 Department of Mental Health Research, National Center for Mental Health, Seoul, Republic of Korea, 2 Department of Urology, Kyung Hee University, Seoul, Republic of Korea, 3 Department of Psychiatry, Kangwon National University, School of Medicine, Chunchon, Republic of Korea, 4 College of Nursing, Woosuk University, Wanjoo, Republic of Korea, 5 Department of Preventive Medicine, School of Medicine, Kyung Hee University, Seoul, Republic of Korea

* kachas@naver.com

**Data Availability Statement:** We used the National Health Insurance Service- National Sample Cohort (NHIS-NSC) database in South Korea. These data are available to researchers. It could be accessed at

## Abstract

Benign prostatic hyperplasia is a commonly diagnosed disease in elderly men, but elderly men with benign prostatic hyperplasia are more likely to have a lower quality of life and depressive symptoms. This study aims to examine the association benign prostatic hyperplasia patients with suicide death relative to a control group comprising individuals without benign prostatic hyperplasia. We used the Korean National Health Insurance Service-National Sample Cohort from 2006 to 2015 comprising of 193,785 Korean adults ≥40 years old, and followed-up for suicide death during the 8.7 years period. Cox-proportional hazard model was used to estimate hazard ratios for suicide among patients with benign prostatic hyperplasia. From 2006 to 2010, a total of 32,215 people were newly diagnosed with benign prostatic hyperplasia. The suicide rate of people without benign prostatic hyperplasia was 61.6 per 100,000 person-years, whereas that of patients with benign prostatic hyperplasia was 97.3 per 100,000 person-years, 1.58 times higher than the control group (p<0.01). After adjusting for covariates, the hazard ratio for suicide among patients with benign prostatic hyperplasia was 1.47 (95% C.I. = 1.21 to 1.78; p<0.01) compared to people without benign prostatic hyperplasia. For men without mental disorders, the hazard ratio for suicide among patients with benign prostatic hyperplasia was 1.36 (95% CI = 1.05 to 1.76) compared to control group after adjusting for multiple covariates. Our study suggests that men with benign prostatic hyperplasia had a higher probability of suicide compared to men without benign prostatic hyperplasia in South Korea. This study suggests that physicians may be aware that men newly diagnosed with benign prostatic hyperplasia had high probability of suicide.

## Introduction

Benign prostatic hyperplasia (BPH) is a common disease in men aged≥40 years old. According to the Olmsted County study, the proportion of patients with a moderate to severe composite of obstructive symptoms increased from 13% in men aged 40–49 years to 28% in men

https://nhiss.nhis.or.kr/bd/ab/bdaba002cv.do.
However, it is not open for free, and researchers
have to pay a certain amount for use. This
database also cannot be taken out freely and must
be accessed using a virtual computer system. We
have no special privileges in accessing the data
from NHIS-NSC.

**Funding:** This study was supported by grant of the
National Center for Mental Health (2018-08). The
first author(Sang-Uk Lee) received NCMH grant.
The sponsors played no role in study design, data
collection, analysis and interpretation of study
findings.

**Competing interests:** NO authors have competing
interests.

aged >70 years [1]. Debra et al. estimated that about 1.9 billion adults aged ≥20 years experienced symptoms associated with lower urinary tract obstruction worldwide in 2008 [2]. In South Korea, the estimated number of patients with BPH has continuously increased from 894,908 in 2012 to 1,191,595 in 2017 [3]. The number of patients with BPH is expected to increase with the aging population, sedentary life style and increasing obesity.

Major depression is well-known risk factor for suicide [4]. Some epidemiological studies have reported that lower urinary tract symptoms (LUTS), and sexual dysfunction, which are closely related to BPH, contribute to increased probability of depressive symptoms and lower the quality of life (QOL) of patients. Indeed, a cross-sectional study reported that moderate to severe LUTS is associated with decreased QOL in patients with prostate disease [5]. A few studies have also demonstrated that men with LUTS are more likely to have depressive symptoms or suicidal ideation [6–9]. Pietrzyk et al. reported that 22.4% of people treated with BPH had depressive symptoms and prevalence of depressive symptoms were associated with the severity of LUTS, BPH therapy and erectile dysfunction [10]. In United States, Breyer et al. also showed that the odds ratio for suicidal ideation among men with ≥2 LUTS was 3.5 times higher than the control group (≤1 LUTS) [6]. In addition, it was suggested that the 5alpha-reductase inhibitors (5aRI) could lower the quality of life of men with BPH.

However, a recent large retrospective cohort study demonstrated that the use of 5aRI was associated with increased risk of depression or self-harm, but did not increase the risk of suicide [11]. Besides, all men with depression or suicide ideation do not necessarily advance to suicide [12]. To the best of our knowledge, there are barely any studies published examining the association between BPH and risk of suicide death itself.

Therefore, it is necessary to examine whether men diagnosed with BPH have increased probability of suicide death in a real community setting. The present study aimed to evaluate the suicide mortality of Korean men diagnosed with BPH using nationwide representative cohort data of 193,785 men aged ≥40 years.

## Material and methods

### Study population

This was a retrospective cohort study using the National Health Insurance Service- National Sample Cohort (NHIS-NSC) database. The NHIS-NSC database was obtained from the National Health Insurance Corporation to examine the probability of suicide mortality among patients with newly diagnosed BPH [13]. Everyone living in South Korea is obliged to participate in the National Health Insurance Service. The NHIS-NSC database comprises of data obtained using stratified random sampling based on sex, age, income level and region from the original database of the National Health Insurance Service database and the medical aid database. The NHIS-NSC database includes patients' information such as age, sex and health insurance cost, health examination data and medical records such as prescription, surgery, disease diagnosis and hospital information. The NHIS-NSC database is also linked to the cause and date of death obtained from Statistics Korea. The last follow-up date for death was December 31, 2015.

### Definition of benign prostatic hyperplasia

Patients with BPH were defined by having the primary code "N40" according to the ICD-10 codes [14]. In order to restrict the study population to incident cases (newly diagnosed cases), the washing-out period was set from January 1 2002 to December 31 2005. patients already diagnosed with BPH before year 2006 were excluded and only patients with newly diagnosed with BPH were included from 2006 to 2015.

## Suicide mortality

The primary endpoint was death from suicide. Death from suicide was defined by the code "X60-X84" according to the ICD-10 codes [15, 16].

## Ethics statement

The study protocol was approved by the Institutional Review Board of National Center for Mental Health (IRB No.116271-2018-32). Our study protocol was exempted from review because it was not a clinical trial, which required contacting the patients, and NHIS-NSC database was anonymized to prevent identification of individuals.

## Selection of the study participants

The NHIS-NSC database represents 2.2% of all Korean adults in 2002 who were followed up to 2015.13 We used the NHIS-NSC data from 2006 to 2015, because, there were many missing data points for health insurance premiums before the year 2006. A total of 1,021,208 participants were included in the NHIS-NSC database from 2006 to 2015 (Fig 1).

Of these, 556,724 participants were excluded, because they were younger than 40 years; 241,315 women were also excluded; 9,914 patients diagnosed or treated for BPH in 2006 were

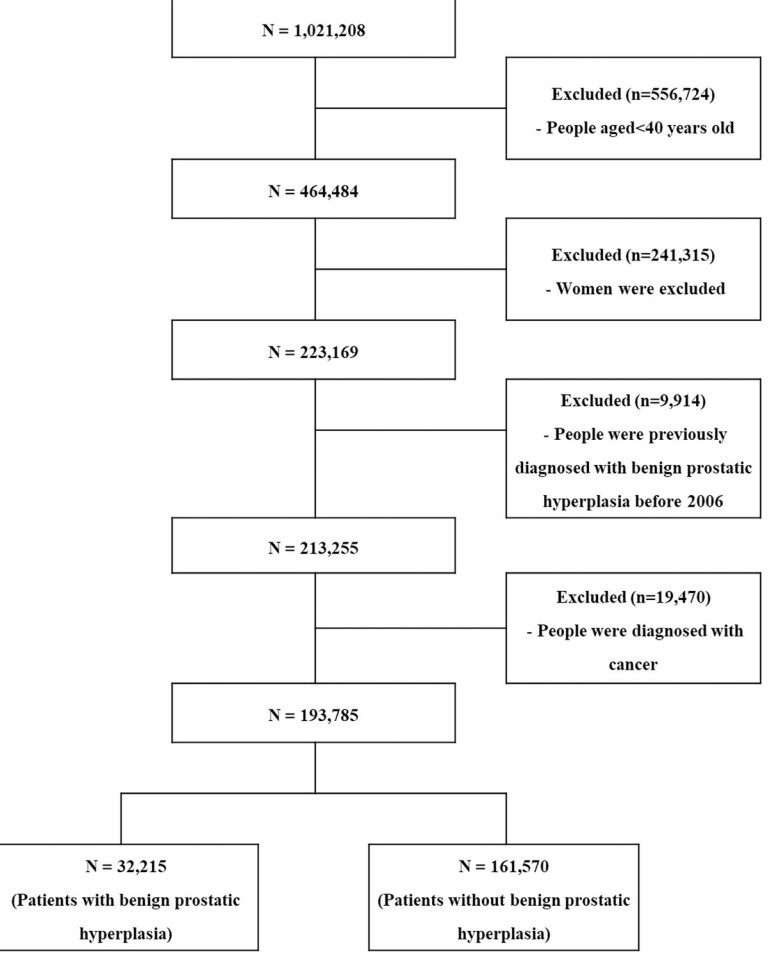

**Fig 1. Flow-chart for selection of study participants.**

excluded, because our study focused only on cases with newly diagnosed BPH. Consequently, 19,470 cancer patients were excluded, because the quality of life of cancer patients is worse than that of general population and cancer patients have a higher risk of suicide. Lastly, 193,785 participants who were not diagnosed with BPH prior to 2007 were included in the final study population.

## Covariates

The region was further categorized into cities and provinces according to administrative districts [16]. The income level was classified based on the health insurance premium, and it was divided into high income (9th-10th quartiles of the premiums), middle income (5th-8th quartiles of the premiums), and low income (1st-4th quartiles of the premiums) levels and the medical aid group [16]. Disability was categorized as "People without disability" and "People with disability" according to the disability criteria set by the Ministry of Health and Welfare, and past medical diagnosis by a physician [16]. Co-morbidity was defined among people who had more than one of the 41 pertinent disease categories [17]. The presence of mental disorders was defined by diagnosis with the code "F00-F99" from 2006 to 2015. Unfortunately, the NHIS-NSC considered detailed information about mental disorders as sensitive information and did not provide sub-code of mental disorders to general researchers.

## Statistical analysis

In this study, baseline characteristics of patients diagnosed with BPH and the population without BPH were compared using independent t-test for continuous variables and chi-square test for categorical variables. The Cox-proportional hazard model was used to examine the hazard ratio for suicide risk among patients newly diagnosed with BPH compared to the population without BPH. In order to adjust confounding bias, age, region and number of comorbidities were adjusted in model 1 and age, region, number of comorbidities, disabilities, mental disorders and income level were adjusted in model 2. In addition, we conducted stratified analysis by age group (40–59 years/60-69 years/≥70 years) and income level (medical aid/low income level/middle income level/high income level) to examine if there were differences in the association between BPH and rates of suicide by age, income level and previous history of mental health disorder. Interaction effects of age, income level and BPH on risk of suicide were tested using the likelihood ratio test [18]. We performed post-hoc power calculation using the hazard ratio of the Cox-proportional hazard model and the proportion for suicide mortality of BPH group and control group using "powerSurvEpi" package of R program.

P-values less than 0.05 denoted statistical significance. SAS Enterprise Guide ver. 6.1 (SAS Institute Inc., Cary, NC, USA) and R 3.4.0 (R Foundation for Statistical Computing, Vienna, Austria) were used for all statistical analyses.

## Results

### Baseline characteristics of study participants

The baseline characteristics of patients with BPH and the general population without BPH are presented in Table 1. During the average 8.7-years of follow-up, 32,215 adults aged ≥40 years were newly diagnosed with BPH. Mean age of study participants were 53.2±10.5 years old and all participants were Korean population. Patients with BPH were much older and had higher income level, higher proportion of disabilities and mental disorders compared to those without BPH.

**Table 1. Baseline characteristics of patients diagnosed with benign prostatic hyperplasia and general population without benign prostatic hyperplasia.**

| Characteristics | Overall | People with benign prostatic hyperplasia | People without benign prostatic hyperplasia | p-value[a] |
|---|---|---|---|---|
| | Number (%) or mean (SD) | Number (%) or mean (SD) | Number (%) or mean (SD) | |
| Total number | 193,785 (100.0%) | 32,215 (100.0%) | 161,570 (100.0%) | |
| Age (years) | 53.2 (10.5) | 59.0 (10.4) | 52.1 (10.2) | <0.0001 |
| Region | | | | |
| City | 89,243 (46.1%) | 15,250 (47.3%) | 73,993 (45.8%) | <0.0001 |
| Province | 104,542 (53.9%) | 16,965 (52.7%) | 87,577 (54.2%) | |
| Income level | | | | |
| Medical aid | 8182 (4.2%) | 1513 (4.7%) | 6669 (4.1%) | <0.0001 |
| Low | 54173 (27.9%) | 8427 (26.2%) | 45746 (28.3%) | |
| Middle | 75888 (39.2%) | 11673 (36.2%) | 64215 (39.8%) | |
| High | 55542 (28.7%) | 10602 (32.9%) | 44940 (27.8%) | |
| Disability | | | | |
| No disability | 179395 (92.6%) | 29247 (90.8%) | 150148 (92.9%) | <0.0001 |
| People with disability | 14390 (7.4%) | 2968 (9.2%) | 11422 (7.1%) | |
| Mental disorder | | | | |
| No mental disorder | 168802 (87.1%) | 24760 (76.9%) | 144042 (89.2%) | <0.0001 |
| People with mental disorder | 24983 (12.9%) | 7455 (23.1%) | 17528 (10.8%) | |
| Number of comorbidities | 3.71 (2.8) | 5.02 (3.1) | 3.45 (2.6) | <0.0001 |

Continuous variables are presented as mean (standard deviation) and categorical variables are presented as numbers (percentage).

[a]Independent t-test for continuous variables and chi-square test for categorical variables were used to test difference between men with BPH and men without BPH.

During the follow-up period, we found 1,006 cases of suicide death among a total of 193,785 men (Table 2). Patients with BPH had a suicide rate of 97.3 (95% CI = 82.4 to 114.0) per 100,000 person-years, which was about 1.6 times higher than the suicide rate of those without BPH (61.6 (95% CI = 57.6 to 65.8) per 100,000 person-years).

## The hazard ratio for suicide associated with benign prostatic hyperplasia

The cox-proportional hazard model was used to examine the hazard ratio for suicide among patients with BPH compared to general population without BPH after adjusting for covariates (Table 3). In the unadjusted model, hazard ratio (HR) for suicide among patients with BPH was 1.47 (95% CI = 1.23 to 1.76) compared to general population without BPH. After adjusting for age, geographical location, and comorbidities, patients with BPH had a higher hazard ratio for suicide [HR: 1.54 (95% CI = 1.28 to 1.86)] than general population without BPH. In the final model, the hazard ratio (HR) for suicide among patients with BPH was 1.47 (95%

**Table 2. Number of suicide death and suicide rates in patients diagnosed with benign prostatic hyperplasia and general population without benign prostatic hyperplasia.**

| Categories | Total number | Number of suicide death | Suicide rate per 100,000 person-years (95% CI)[a] |
|---|---|---|---|
| Patients with benign prostatic hyperplasia | 32,215 | 146 | 97.3 (95% CI = 82.4 to 114.0) |
| General population without benign prostatic hyperplasia | 161,570 | 860 | 61.6 (95% CI = 57.6 to 65.8) |

CI: Confidence interval.

[a]Suicide rates are expressed as incidence density per 100,000 person-years.

**Table 3. Hazard ratios (95% CI) for suicide among patients with benign prostatic hyperplasia compared to general population without benign prostatic hyperplasia.**

| Variables | Unadjusted HR (95% CI) | Age, region and comorbidity adjusted HR (95% CI) | Multivariable adjusted HR (95% CI)[a] |
|---|---|---|---|
| **Benign prostatic hyperplasia** | | | |
| No | 1.00 (Reference) | 1.00 (Reference) | 1.00 (Reference) |
| Yes | 1.47 (1.23 to 1.76) | 1.54 (1.28 to 1.86) | 1.47 (1.21 to 1.78) |
| Age (years) | | 1.05 (1.05 to 1.06) | 1.04 (1.04 to 1.05) |
| Region | | | |
| City | | 1.00 (Reference) | 1.00 (Reference) |
| Province | | 1.16 (1.02 to 1.31) | 1.12 (0.99 to 1.28) |
| Number of Comorbidity | | 0.84 (0.82 to 0.86) | 0.82 (0.80 to 0.84) |
| Income level | | | |
| Medical aid | | | 1.83 (1.40 to 2.39) |
| Low | | | 1.56 (1.31 to 1.85) |
| Middle | | | 1.30 (1.10 to 1.53) |
| High | | | 1.00 (Reference) |
| Disability | | | |
| No | | | 1.00 (Reference) |
| Yes | | | 1.25 (1.02 to 1.54) |
| Mental disorder | | | |
| No | | | 1.00 (Reference) |
| Yes | | | 3.26 (2.82 to 3.77) |

CI: Confidence interval.

[a]Multivariable model was adjusted for age, geographical location, comorbidities, disabilities, mental health, and income level.

CI = 1.21 to 1.78) compared general population without BPH after adjusting for age, geographical location, comorbidities, disabilities, mental health, and income level. The post-hoc power was 99.9% when the post-hoc power calculation was performed using the hazard ratio for suicide and the proportion for suicide of BPH group and control group.

### The hazard ratio of suicide associated with benign prostatic hyperplasia by age groups

We performed subgroup analysis to evaluate whether there was as significant difference in HR by age group, because the age was a major confounding variable with the greatest effect on risk of suicide and BPH (Table 4). Among the age group of 40–59 years, HR for suicide among men with BPH was 1.64 times higher (95% CI = 1.20 to 2.22) than that in general population without BPH; the HR for suicide among men with BPH aged more than 60 years was 1.72 times higher (95% CI = 1.24 to 2.40) compared to the control group. However, there was no significant difference in the suicide risk between men with BPH and men without BPH in the >70 years old group.

### The hazard ratio of suicide associated with benign prostatic hyperplasia by income level groups

Subgroup analysis was also performed by income levels, because the socioeconomic status is a major health determinant of suicide and BPH (S1 Table). The Cox-proportional hazard model was used in all groups after adjusting for all covariates. Although there was no significant interaction between income level and BPH, there was a small difference in the association between

**Table 4. Hazard ratios (95% CI) for suicide among patients with benign prostatic hyperplasia according to different age groups.**

| Characteristics | Suicide rates (95% CI) | Multivariable adjusted HR (95% CI)[a] | p for interaction[b] |
|---|---|---|---|
| 40–59 years old | | | 0.78 |
| Without benign prostatic hyperplasia | 49.7 (45.7 to 53.9) | 1.00 (reference) | |
| With benign prostatic hyperplasia | 64.0 (47.9 to 83.9) | 1.64 (1.20 to 2.22) | |
| 60–69 years old | | | |
| Without benign prostatic hyperplasia | 92.0 (78.7 to 107.0) | 1.00 (reference) | |
| With benign prostatic hyperplasia | 116.0 (88.0 to 150.2) | 1.72 (1.24 to 2.40) | |
| ≥ 70 years old | | | |
| Without benign prostatic hyperplasia | 157.8 (132.6 to 186.5) | 1.00 (reference) | |
| With benign prostatic hyperplasia | 159.3 (116.7 to 212.5) | 1.11 (0.77 to 1.60) | |

Suicide rates are expressed as incidence density per 100,000 person-years.

[a]Multivariable model was adjusted for age, geographical location, comorbidities, disabilities, mental health, and income level.

[b]P for interaction was tested using likelihood ratio test.

BPH and risk of suicide according to the income level. There was no difference in the risk of suicide in patients with BPH compared to general population without BPH in the medical aid and low income level groups. However, men with BPH had a higher risk of suicide compared to men without BPH in middle and high income groups.

### The hazard ratio of suicide associated with benign prostatic hyperplasia by mental health disorder

Mental disorders, especially those related to major depression are a well-known risk factor for suicide (S2 Table). Therefore, we also examined the risk of suicide due to BPH by the presence of the mental disorders. Among men without mental disorders, patients with BPH had about 1.36 (95% CI = 1.05 to 1.76) higher risk of suicide compared to men without BPH after adjusting for multiple covariates. Among men with previous mental disorders, the hazard ratio for men with BPH was 1.66 (95% CI = 1.24 to 2.21) compared to men without BPH using a fully adjusted cox-proportional hazard model.

## Discussion

Our findings show that the probability of suicide among patients diagnosed with BPH was about 1.5 times higher than those not diagnosed with BPH. The association between BPH and suicide was similar (HR = 1.66 (95% CI = 1.24 to 2.21)) even in the group without mental disorder. Although previous studies have demonstrated that patients diagnosed with BPH or having LUTS have more depressive symptoms and poorer quality of life compared to controls, the fact that this leads to suicide is rarely known based on our knowledge.

### The association between BPH and suicide

The causes of suicide are highly complex ranging from internal factors to external ones, including involvement of genetic, psychological, socio-cultural factors as well as personal experiences like trauma [19]. However, factors such as depression or economic crisis are well-known to increase the risk of suicide [20, 21]. In addition, there is growing evidence that patients diagnosed with diseases that lower the quality of life such as, prostate cancer and asthma also increase the probability of suicide [22, 23].

However, there were very few studies which have focused on the nature and direction of relationship between BPH and depression. In addition, there are rare studies on the direction and association of BPH, depression, suicide. Interestingly, our study findings showed that patients with BPH have high probability of suicide, regardless of depressive symptoms. Therefore, we were able to infer the two directions from previous study findings and discussion. At first, patients were newly diagnosed with BPH will develop depressive symptoms, and these depressive symptoms may aggravate lower urinary tract symptoms and cause a vicious cycle that aggravates depression and finally leads to suicide [8]. The second hypothesis is that the risk of BPH or LUTS is increased by the dysregulation of the hypothalamic–pituitary–adrenal (HPA) axis due to the predisposing depressive symptoms [24]. People who committed suicide without a diagnosis of depression actually had depressive symptom due to BPH/LUTS but they may don't want to visit to the psychiatric clinic. Or it is possible that they decided to commit suicide immediately due to uncontrolled or worsening LUTS (Erectile dysfunction, nocturia, urinary dysfunction etc.).

## Other biological relevance supporting the association between BPH and suicide

BPH leads to lower urinary tract disease, impairs sexual function, decreases the quality of life and increases the risk of depression or depressive symptoms [5–10, 25]. From this perspective, it was not surprising that patients diagnosed with BPH had a higher probability of suicide compared to control group without BPH. Recent studies have reported that impairment of serotonin (5-HT) synthesis was associated with both urinary dysfunction and depression, suggesting common pathophysiological mechanisms in lower urinary tract symptoms and depression [26, 27]. In addition, some studies have shown that increased adrenergic tone was associated with depressive and voiding symptoms [28] and the corticotropin releasing factor (CRF) pathway leads to changes in stress-related depressive symptoms as well as urinary symptoms [29]. In addition, a recent randomized controlled trial showed that additional anti-depressant therapy not only improved depression, but also reduced symptoms of LUTS and improved quality of life [30]. These pathophysiological studies also support that BPH or its associated lower urinary tract symptoms can increase the risk of depression, leading to suicide.

## Strengths and limitations

Some limitations should be considered while interpreting our study's findings. First, our study design could not account for the severity of BPH, because the NHIS-NSC database was based on secondary data for estimating payments for health insurance. The NHIS-NSC database did not include severity or symptoms of patients with BPH. Second, we defined patients with BPH as those having the code "N40" according to the ICD-10 codes. However, some up-coding or misclassification for the diagnosis for prostatic hyperplasia may exist. Although the use of ICD-10 code "N40" has the potential for misclassification of BPH, it is often used to estimate the national incidence or prevalence of BPH [31]. Third, covariates such as co-morbidities, mental disorders and income levels were adjusted in the Cox-proportional hazard model to minimize the effects of confounding variables. We also performed stratification analysis to exclude the effects of major confounding variables. However, there may be residual confounding factors for suicide such as personal experiences of trauma, cultural factors or severe stress which we could not adjust in this study. Fourth, different types of LUTS may be more likely to be associated with mental health disease [32], however, we could not evaluate the effects of LUTS on depression or suicide in our study. Fifth, 5α-reductase inhibitors, which were mainly used in the medical treatment of BPH, were not reported previously to increase the risk of

depression [11], but it may be necessary to consider the effects of medication to more accurately investigate the association between BPH and probability of suicide. Finally, our study findings cannot be easily generalized to other countries. South Korea is considered as a developed country with a very rapid economic growth [33]. At the same time, South Korea has the highest suicide rate among OECD countries [15]. While South Korea has the advantage of physically and economically easy access to medical service, there are cultural barriers that are reluctant to visit psychiatry for mental disorder [34].

Despite these limitations, our study showed meaningful findings that patients with BPH have a higher probability of suicide compared to general population using a large sample cohort data representing South Korea, especially considering the fact that there are barely any studies show an association between BPH and probability of suicide.

## Clinical implications

Patients with BPH may complain of sexual dysfunction and dysuria and had lower quality of life. There has been a lot of clinical evidence that these problems among patients with BPH threaten the quality of life, and even lead to depression. Our study findings show that patients with BPH are higher risk for suicide compared to general population and some of them would be necessary for mental health care to prevent suicide. Primary care physicians may have more attention to not only the voiding symptoms related to BPH, but also mood disorders such as anxiety and depression.

## Conclusion

Our study suggested that probability of suicide was 1.5 times higher in men with BPH compared to men without BPH among those aged over 40 years in Korea. This close association between BPH and suicide was observed even in people without mental disorder. Further studies are warranted to know high risk group on which BPH patients are at risk for suicide and whether they need transfer and management of mental health.

## Supporting information

**S1 Table. Hazard ratios (95% CI) for suicide among patients with benign prostatic hyperplasia according to different income levels.**
(DOC)

**S2 Table. Hazard ratios (95% CI) for suicide among patients with benign prostatic hyperplasia according to presence of mental disorders.**
(DOC)

## Acknowledgments

We used the National Health Insurance Service–National Sample Cohort (NHIS-NSC) database and the dataset was obtained from the National Health Insurance Service. Our study findings were not related to the National Health Insurance Service.

## Author Contributions

**Conceptualization:** Sang Hyub Lee, Jong-Ik Park, Chang-Mo Oh.

**Data curation:** Sang-Uk Lee.

**Formal analysis:** Sang-Uk Lee.

**Funding acquisition:** Sang-Uk Lee.

**Investigation:** Sang-Uk Lee, Ah-Hyun So, Jong-Ik Park, Soojung Lee, In-Hwan Oh.

**Methodology:** Sang-Uk Lee.

**Project administration:** Chang-Mo Oh.

**Supervision:** Chang-Mo Oh.

**Validation:** Sang Hyub Lee, Ah-Hyun So, Jong-Ik Park, Soojung Lee, In-Hwan Oh.

**Writing – original draft:** Sang-Uk Lee.

**Writing – review & editing:** Sang Hyub Lee, Ah-Hyun So, Jong-Ik Park, Soojung Lee, In-Hwan Oh, Chang-Mo Oh.

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
