## [Decision Letter · Decision Letter 0]

10 Sep 2021

PONE-D-21-21842

Risk of Suicide among Patients with Benign Prostatic Hyperplasia in South Korea: a Nationwide Retrospective Cohort Study

PLOS ONE

Dear Dr. Oh,

Thank you for submitting your manuscript to PLOS ONE. After careful consideration, we feel that it has merit but does not fully meet PLOS ONE’s publication criteria as it currently stands. Therefore, we invite you to submit a revised version of the manuscript that addresses the points raised during the review process.

ACADEMIC EDITOR:

Can you with help of the reviewers comments revise thae manuscript. Can you also explain in the manuscript what usually leads th coding BPH, is that 'having lower urinary tract symptoms' or is that based on measuring the size of the prostate. You say in your 'limitations' that you do not know exactly per patient, but you can refer to (a) national (Urologists? GP-PCP?) standard(s). Also can the comparator group can have enlarged prostate without the coding because the prostate was never measured? I also find some (more) reports on relevant medication (5AR and alphablocker) and suicide risk, can you include this? Can you also define 'newly diagnosed' and discuss this in the perspective of your follow up. (and better explain how you excluded patients that were (how?) treated during that follow up).

We look forward to receiving your revised manuscript.

Kind regards,

Peter F.W.M. Rosier, M.D. PhD

Academic Editor

PLOS ONE

“This study was supported by grant of the National Center for Mental Health (2018-08). The sponsors played no role in study design, data collection, analysis and interpretation of study findings.”

Please note that funding information should not appear in the other areas of your manuscript. We will only publish funding information present in the Funding Statement section of the online submission form.

“This study was supported by grant of the National Center for Mental Health (2018-08). The first author (Sang-Uk Lee) received NCMH grant. The sponsors played no role in study design, data collection, analysis and interpretation of study findings.”

“NO authors have competing interests”

Reviewers' comments:

Reviewer's Responses to Questions

**Comments to the Author**

1. Is the manuscript technically sound, and do the data support the conclusions?

Reviewer #1: Partly

Reviewer #2: Partly

2. Has the statistical analysis been performed appropriately and rigorously? 

Reviewer #1: Yes

Reviewer #2: Yes

3. Have the authors made all data underlying the findings in their manuscript fully available?

Reviewer #1: Yes

Reviewer #2: Yes

4. Is the manuscript presented in an intelligible fashion and written in standard English?

Reviewer #1: Yes

Reviewer #2: Yes

5. Review Comments to the Author

Reviewer #1: The important massage from the study is that patients with BPH and without history of mental disorders have increased risk of suicide. This should be presented in the abstract and perhaps in the title.

Please consider a subanalysis after exclusion of subjects with mental disorders.

Surprisingly the analysis shows that the number of comorbidities is increasing the risk for suicide. Perhaps the authors should look for the 41 disease categories and identified the risk factors. In addition it would be wise to analyze multimorbidity as a risk factor.

Introduction

Pietrzyk et al. do not present suicidal ideation. They demonstrate that occurrence of depressive symptoms in patients with BPH is independently associated with severity of LUTS [OR 1.1 (95%CI: 1.09-1.13), BPH therapy (polytherapy, history of TURP), erectile dysfunction and comorbidities.

Discussion

Discussion need to be improved incorporating new data from additional analyses.

The authors extensively discuss biological relevance supporting the association between BPH and risk of suicide, but they do not have anything to add in this point. This part should be more consist.

Clinical implications are nonrealistic. I can’t believe that urologists can be more holistic. But it might be a place for general practitioners to look for depressive symptoms in their patients treated for BPH/LUTS.

The conclusion should be based on the risk in patients without mental disorders.

Reviewer #2: Congratulation to the authors to look into the relation between benign prostatic hyperplasia and suicide. Due to it is a retrospective study, the title and conclusion should be suggestive of suicide and not risk of suicide.

6. PLOS authors have the option to publish the peer review history of their article (what does this mean?). If published, this will include your full peer review and any attached files.

Reviewer #1: No

Reviewer #2: No

---

## [Author Response · Author response to Decision Letter 0]

27 Oct 2021

Manuscript ID: PONE-D-21-21842

Original title: 

Risk of Suicide among Patients with Benign Prostatic Hyperplasia in South Korea: a Nationwide Retrospective Cohort Study

Revised title: 

Association between Benign Prostatic Hyperplasia and Suicide in South Korea: a Nationwide Retrospective Cohort Study

To the Editors and Reviewers: 

We thank you for careful reading of the manuscript and helpful comments and suggestions from reviewers. We are submitting reviewers’ comments together, and plan how we revised our paper according to the reviewers’ suggestions and comments. Changes made to the text are marked with highlights changes and track change mode in the revised manuscript.

Comments from Academic Editor: 

1. Can you with help of the reviewers’ comments revise the manuscript? Can you also explain in the manuscript what usually leads th coding BPH, is that 'having lower urinary tract symptoms' or is that based on measuring the size of the prostate?

Answer: Thanks for your thoughtful comments. We believe that we have faithfully answered the reviewers' questions, and reviewer and editor’s comments have improved the quality of our research.

As the Editor pointed out, the definition of benign prostatic hyperplasia (BPH) is a very important issue for this study. Generally, the diagnosis of lower urinary tract symptoms (LUTS)/benign prostatic hyperplasia (BPH) is established by the presence of storage, voiding, and/or irritative urinary symptoms in the absence of history, examination or laboratory findings suggesting of non-BPH causes of lower urinary tract symptoms. 

However, we used data from the Health Insurance Corporation, which is a secondary source of data. Although ICD-10 code for BPH (“N40”) used as the main diagnosis may have some degree of misclassification, it is believed to have a considerable degree of validity for real (true) lower urinary tract symptoms (LUTS)/benign prostatic hyperplasia (BPH).

Indeed, the previous studies using data from the National Health Insurance Corporation also defined that BPH as ICD-10 code “N40”. The national incidence rate for benign prostatic hyperplasia (BPH) is also estimated using ICD-10 code “N40”.

Reference) Kim SH, Kwon WA, Joung JY. Impact of Benign Prostatic Hyperplasia and/or Prostatitis on the Risk of Prostate Cancer in Korean Patients. World J Mens Health. 2021;39(2):358-365. doi: 10.5534/wjmh.190135 [doi].

Lee YJ, Lee JW, Park J, Seo SI, Chung JI, Yoo TK, Son H. Nationwide incidence and treatment pattern of benign prostatic hyperplasia in Korea. Investig Clin Urol. 2016;57(6):424-430. doi: 10.4111/icu.2016.57.6.424.

Also, in the national patient registry of denmark, ICD-10 code "N40" was found to have a positive predictive value (PPV) of 95% (95% CI: 89–98%) for benign prostatic hyperplasia (BPH). (Of course, since this case is a disease registry, the accuracy is thought to be higher than our data.)

Reference) Bengtsen MB, Heide-Jørgensen U, Blichert-Refsgaard LS, Hjelholt TJ, Borre M, Nørgaard M. Positive Predictive Value of Benign Prostatic Hyperplasia and Acute Urinary Retention in the Danish National Patient Registry: A Validation Study. Clin Epidemiol. 2020;12:1281-1285. doi: 10.2147/CLEP.S278554.

2. You say in your 'limitations' that you do not know exactly per patient, but you can refer to (a) national (Urologists? GP-PCP?) standard(s).

Answer: There is no established diagnostic standard for benign prostatic hyperplasia (BPH). However, the International Prostate Symptom Score (IPSS) questionnaire was adopted as a basic questionnaire standard at the International Council of BPH organized by the World Health Organization in 1993, and various studies on epidemiology and therapeutic efficacy have been done using the IPSS [7]. The IPSS is used to assess the severity of storage symptoms and voiding symptoms with one additional quality of life question.

However, because our data is consisted of secondary database (the National Health Insurance Service- National Sample Cohort (NHIS-NSC) database) rather than clinical chart review, it is impossible to use IPSS questionnaire or uroflowmetry or measurement of postvoid residual volume (PVR).

However, ICD-10 code of “N40” is often used to measure the nationwide incidence rate for benign prostatic hyperplasia (BPH).

Reference) Kim SH, Kwon WA, Joung JY. Impact of Benign Prostatic Hyperplasia and/or Prostatitis on the Risk of Prostate Cancer in Korean Patients. World J Mens Health. 2021;39(2):358-365. doi: 10.5534/wjmh.190135 [doi].

Lee YJ, Lee JW, Park J, Seo SI, Chung JI, Yoo TK, Son H. Nationwide incidence and treatment pattern of benign prostatic hyperplasia in Korea. Investig Clin Urol. 2016;57(6):424-430. doi: 10.4111/icu.2016.57.6.424.

(Discussion, Page 16, Line 314-315) Although the use of ICD-10 code "N40" has the potential for misclassification of BPH, it is often used to estimate the national incidence or prevalence of BPH [31].

31. Lee YJ, Lee JW, Park J, Seo SI, Chung JI, Yoo TK, Son H. Nationwide incidence and treatment pattern of benign prostatic hyperplasia in Korea. Investig Clin Urol. 2016;57(6):424-430. doi: 10.4111/icu.2016.57.6.424.

3. Also can the comparator group can have enlarged prostate without the coding because the prostate was never measured?

Answer: Of course, there is a probability that some very small number of control groups have enlarged prostate. Some patients with mild BPH may don’t want to go hospital due to fear of treatment or financial difficulties. However, it is practically impossible to find such a case with hospital patient-control study. BPH is very difficult to detect without a doctor's diagnosis, unless there was a large survey using IPSS questionnaire.

4. I also find some (more) reports on relevant medication (5AR and alphablocker) and suicide risk, can you include this? 

Answer: Whether 5-a reductase or alpha blocker increases the risk of suicide in patients with benign prostatic hyperplasia (BPH) is a very important and controversial issue [1-3]. However, our study has a slightly different scope from this topic (the association between medication (5AR and alphablocker) and suicide risk). 

At first, the control group in our study was the general population without a diagnosis of BPH. However, to evaluate the association between medication (5AR and alphablocker) and suicide risk, the control group must also be patients with BPH. This is, because BPH itself can reduce sexual dysfunction and quality of life due to its symptoms.

Second, we do not have information about the medication (5AR and alphablocker). Of course, the National Health Insurance Service- National Sample Cohort (NHIS-NSC) database has information about the medication (5AR and alphablocker). If we ask NIHS to reuse the data for further analysis (information on the medication), we may have to wait to use after one year. So, we would be very grateful if the editor could understand our situation.

Perhaps, we cannot be completely excluded the effect of medication such as 5AR or alpha blocker on the increased risk of suicide among people newly diagnosed with benign prostatic hyperplasia. However, it was not possible the effects of medication on the risk of suicide in our study finding.

Moreover, we thought that depressive symptoms and BPH are interactively influenced by each other [4-5] (bidirectional relationship), it is difficult to say that the increased risk of suicide in patients with BPH in our study is simply due to medication (5AR and alphablocker).

Reference)

1. Irwig MS. Depressive symptoms and suicidal thoughts among former users of finasteride with persistent sexual side effects. J Clin Psychiatry. 2012;73(9):1220-3.

2. Welk B, McArthur E, Ordon M, Anderson KK, Hayward J, Dixon S. Association of suicidality and depression With 5alpha-Reductase Inhibitors. JAMA Intern Med. 2017;177:683-91.

3. Nguyen DD, Marchese M, Cone EB, Paciotti M, Basaria S, Bhojani N, Trinh QD. Investigation of Suicidality and Psychological Adverse Events in Patients Treated With Finasteride. JAMA Dermatol. 2021;157(1):35-42.

4. Dunphy C, Laor L, Te A, Kaplan S, Chughtai B. Relationship between depression and lower urinary tract symptoms secondary to benign prostatic hyperplasia. Rev Urol. 2015;17:51-7.

5. Huang CL, Wu MP, Ho CH, Wang JJ. The bidirectional relationship between anxiety, depression, and lower urinary track symptoms: A nationwide population-based cohort study. J Psychosom Res. 2017;100:77-82.

5. Can you also define 'newly diagnosed' and discuss this in the perspective of your follow up. (and better explain how you excluded patients that were (how?) treated during that follow up).

Answer: Our focus was on people newly diagnosed with BPH. We were concerned that if people with existing BPH (prevalent cases) were included in this study, a selective survival bias would occur, in which those diagnosed with BPH who had already committed suicide were excluded. Therefore, we have set washing period of 4 years from 2002 to 2006. Patients diagnosed with or treated for BPH during the washing out period were excluded from the study participants, and the starting point was from January 1, 2006. From January 1, 2006 to December 31, 2015 (during study period), patients newly diagnosed with BPH were classified as BPH patients, and the rest were classified as control group. Please see the below figure.

We also slightly revised the method part to make it easier for editors and readers to understand as follows:

(Method, Page 6, Line 107-111) Patients with BPH were defined by having the primary code “N40” according to the ICD-10 codes [14]. In order to restrict the study population to incident cases (newly diagnosed cases), the washing-out period was set from January 1 2002 to December 31 2005. patients already diagnosed with BPH before year 2006 were excluded and only patients with newly diagnosed with BPH were included from 2006 to 2015 (Figure 1).

Comments from Reviewer #1: 

Reviewer #1: 

1. The important message from the study is that patients with BPH and without history of mental disorders have increased risk of suicide. This should be presented in the abstract and perhaps in the title.

Answer: Thanks for your thoughtful comments. As reviewer has commented, we added the association between BPH and suicide among people without mental disorder in the abstract as follows:

(Page 3, Abstract, Line 52-54) For men without mental disorders, the hazard ratio for suicide among patients with benign prostatic hyperplasia was 1.36 (95% CI=1.05 to 1.76) compared to control group after adjusting for multiple covariates.

2. Please consider a subanalysis after exclusion of subjects with mental disorders.

Answer: We also examined the risk of suicide due to BPH by the presence of the mental disorders. Among men without mental disorders, patients with BPH had about 1.36 (95% CI=1.05 to 1.76) higher risk of suicide compared to men without BPH after adjusting for multiple covariates. Please see the supplementary table 2.

Supplementary Table 2. Hazard ratios (95% CI) for suicide among patients with benign prostatic hyperplasia according to presence of mental disorders

3. Surprisingly the analysis shows that the number of comorbidities is increasing the risk for suicide. Perhaps the authors should look for the 41 disease categories and identified the risk factors. In addition it would be wise to analyze multimorbidity as a risk factor.

Answer: We totally agree with the opinion of reviewers, the number of comorbidities may increase the risk of depression and suicide. Many previous reports and studies have already shown that the presence of mental or physical comorbidities increases the risk of suicidal ideation and suicide [1-4]. One of the reasons for conducting this study is that it has been frequently reported previously that the prevalence of depression among BPH patients is high and that the symptom severity of LUTS is closely related to the prevalence, severity of depressive symptoms, and lower quality of life. However, there have been no reports on the association between BPH and the risk of suicide.

We have requested for permission for remote to the NHIS for revision of the current manuscript, but we heard that there are too many users currently, so we may have to wait for about a year. However, we have excluded 19,470 cancer patients with high suicide risk before the study, and the number of comorbidities and disability were adjusted in multivariate analysis. We would be very grateful if the reviewer could understand our situation.

Reference)

1. Pompili M, Bonanni L, Gualtieri F, Trovini G, Persechino S, Baldessarini RJ. Suicidal risks with psoriasis and atopic dermatitis: Systematic review and meta-analysis. J Psychosom Res. 2021 Feb;141:110347.

2. Alias A, Bertrand L, Bisson-Gervais V, Henry M. Suicide in obstructive lung, cardiovascular and oncological disease. Prev Med. 2021 Nov;152(Pt 1):106543.

3. Bulotiene G, Pociute K. Interventions for Reducing Suicide Risk in Cancer Patients: A Literature Review. Eur J Psychol. 2019 Sep 27;15(3):637-649.

4. Wu JJ, Penfold RB, Primatesta P, Fox TK, Stewart C, Reddy SP, Egeberg A, Liu J, Simon G. The risk of depression, suicidal ideation and suicide attempt in patients with psoriasis, psoriatic arthritis or ankylosing spondylitis. J Eur Acad Dermatol Venereol. 2017 Jul;31(7):1168-1175.

4. (Introduction) Pietrzyk et al. do not present suicidal ideation. They demonstrate that occurrence of depressive symptoms in patients with BPH is independently associated with severity of LUTS [OR 1.1 (95%CI: 1.09-1.13), BPH therapy (polytherapy, history of TURP), erectile dysfunction and comorbidities.

Answer: Thank you for kind and detailed comments. The contents of the bibliography have confused. We changed the sentence pointed out as follows:

(Page 4, Introduction, Line 75-77) Pietrzyk et al. reported that 22.4% of people treated with BPH had depressive symptoms and prevalence of depressive symptoms were associated with the severity of LUTS, BPH therapy and erectile dysfunction [10].

5. (Discussion) Discussion need to be improved incorporating new data from additional analyses. The authors extensively discuss biological relevance supporting the association between BPH and risk of suicide, but they do not have anything to add in this point. This part should be more consist. 

Answer: Thank you for comments. In fact, there was no previous study on the association between BPH and suicide among people without depression. To our best knowledge, this study is the first paper to report the association between BPH and suicide risk. Therefore, it was difficult to develop the contents of the discussion based on our study finding alone. Please understand this difficult point.

As reviewer suggested, we added the following contents to the discussion and conclusion about the increased risk of suicide in people without mental disorders:

(Discussion, page 14 Line 262-264) The association between BPH and suicide was similar (HR=1.66 (95% CI=1.24 to 2.21)) even in the group without mental disorder.

(Discussion, page 14 Line 268-page 15, Line 288) 

(Conclusion, page 17, Line 343-344) This close association between BPH and suicide was observed even in people without mental disorder.

6. Clinical implications are nonrealistic. I can’t believe that urologists can be more holistic. But it might be a place for general practitioners to look for depressive symptoms in their patients treated for BPH/LUTS.

Answer: Thank you for comments. I understood that these parts are the role of the general practitioner or primary care physician rather than the role of urologists. So, we revised the part of clinical implications as follows: 

(Page 17, Clinical implications, Line 337-338) Primary care physicians may have more attention to not only the voiding symptoms related to BPH, but also mood disorders such as anxiety and depression.

7. The conclusion should be based on the risk in patients without mental disorders.

Answer: Thank you for comments. As mentioned above, the absence of a mental disorder is just that people (who maybe have depressive symptoms) has not been diagnosed with an affecting disorder in a mental clinics. People who have attempted suicide may have depressive symptoms, although they have not been diagnosed with a mental disorder in the hospital. 

However, in our study finding, the suicide risk was significantly higher even in people without depression, so we agreed that it should be mentioned in the conclusion as the reviewer has commented.

(Page 3, Abstract, Line 52-54) For men without mental disorders, the hazard ratio for suicide among patients with benign prostatic hyperplasia was 1.36 (95% CI=1.05 to 1.76) compared to control group after adjusting for multiple covariates.

(Conclusion, page 17, Line 343-344) This close association between BPH and suicide was observed even in people without mental disorder.

Comments from Reviewer #2: 

1. Congratulation to the authors to look into the relation between benign prostatic hyperplasia and suicide. Due to it is a retrospective study, the title and conclusion should be suggestive of suicide and not risk of suicide.

Answer: Thank you for comments. We fully agree with the reviewer's opinion. Our study cannot establish a causal relationship, only suggests a possibility between BPH and suicide. Therefore, the title and conclusion have been revised as follows according to the reviewers' opinions:

(Page 1, title): Association between Benign Prostatic Hyperplasia and Suicide in South Korea: a Nationwide Retrospective Cohort Study

(Page 3, Abstract, Line 40-42) This study aims to examine the association benign prostatic hyperplasia patients with suicide death relative to a control group comprising individuals without benign prostatic hyperplasia.

(Page 3, Abstract, Line 54-57) Our study suggests that men with benign prostatic hyperplasia had a higher probability of suicide compared to men without benign prostatic hyperplasia in South Korea. This study suggests that physicians may be aware that men newly diagnosed with benign prostatic hyperplasia had high probability of suicide.

(Page 17, Conclusion, Line 342-343) Our study suggested that probability of suicide was 1.5 times higher in men with BPH compared to men without BPH among those aged over 40 years in Korea.

---

## [Decision Letter · Decision Letter 1]

27 Dec 2021

PONE-D-21-21842R1Association between Benign Prostatic Hyperplasia and Suicide in South Korea: a Nationwide Retrospective Cohort StudyPLOS ONE

Dear Dr. Oh,

Thank you for submitting your manuscript to PLOS ONE. After careful consideration, we feel that it has merit but does not fully meet PLOS ONE’s publication criteria as it currently stands. Therefore, we invite you to submit a revised version of the manuscript that addresses the points raised during the review process.

ACADEMIC EDITOR: Can you add explanation about what the reviewer considers a limitation of the generalisability of the findings?

We look forward to receiving your revised manuscript.

Kind regards,

Peter F.W.M. Rosier, M.D. PhD

Academic Editor

PLOS ONE

Journal Requirements:

Reviewers' comments:

Reviewer's Responses to Questions

**Comments to the Author**

1. If the authors have adequately addressed your comments raised in a previous round of review and you feel that this manuscript is now acceptable for publication, you may indicate that here to bypass the “Comments to the Author” section, enter your conflict of interest statement in the “Confidential to Editor” section, and submit your "Accept" recommendation.

Reviewer #1: All comments have been addressed

Reviewer #3: All comments have been addressed

2. Is the manuscript technically sound, and do the data support the conclusions?

Reviewer #1: Yes

Reviewer #3: Yes

3. Has the statistical analysis been performed appropriately and rigorously? 

Reviewer #1: Yes

Reviewer #3: Yes

4. Have the authors made all data underlying the findings in their manuscript fully available?

Reviewer #1: Yes

Reviewer #3: Yes

5. Is the manuscript presented in an intelligible fashion and written in standard English?

Reviewer #1: Yes

Reviewer #3: Yes

6. Review Comments to the Author

Reviewer #1: The paper was improved. The aswers and modification of the text are acceptable. No further comments.

Reviewer #3: The authors are to be congratulated for the manuscript "Association between Benign Prostatic Hyperplasia and Suicide in South Korea: a Nationwide Retrospective Cohort Study". The previous reviewer comments have been addressed adequately in my opinion.

I have one additional comment.

1) South Korea has one of the world's highest suicide rates per capita. The reasons for this are not clear, but are likely to be multifactorial. As such, the findings in the authors' manuscript may not be generalizable to a different country with lower per capita suicide rates. I would add this as a limitation.

7. PLOS authors have the option to publish the peer review history of their article (what does this mean?). If published, this will include your full peer review and any attached files.

Reviewer #1: No

Reviewer #3: No

---

## [Author Response · Author response to Decision Letter 1]

8 Feb 2022

Manuscript ID: PONE-D-21-21842.R1

Original title: 

Association between Benign Prostatic Hyperplasia and Suicide in South Korea: a Nationwide Retrospective Cohort Study

To the Editors and Reviewers: 

We thank you for careful reading of the manuscript and helpful comments and suggestions from reviewers. We are submitting reviewers’ comments together, and plan how we revised our paper according to the reviewers’ suggestions and comments. Changes made to the text are marked with highlights changes and track change mode in the revised manuscript.

Comments from Academic Editor: 

1. Can you add explanation about what the reviewer considers a limitation of the generalisability of the findings? 

Answer: In my opinion, reviewer is pointing out that we have to consider the specific situation of South Korea – South Korea has the highest suicide rate among OECD country, although South Korea has achieved remarkable success in combining rapid economic growth with significant poverty reduction. There are unique cultural and social background behind this highest suicide rate in South Korea [1-2].

In addition, there is a cultural barrier that are reluctant to visit psychiatry for mental disorder. In particular, there were severe stigma and prejudice toward patients with mental disorders such as depression [3]. In addition, physical illness is common cause for suicide among elderly people [4].

Considering these points, depression due to BPH may be relatively undiagnosed or untreated, and can easily lead to suicide compared to the other countries. These cultural differences (stigma, cultural barrier to visit psychiatry) can lead to difficulties in generalization of our study findings to the situation of other country.

1. Lee SU, Park JI, Lee S, Oh IH, Choi JM, Oh CM. Changing trends in suicide rates in South Korea from 1993 to 2016: a descriptive study. BMJ Open. 2018;8(9):e023144.

2. Hong J, Knapp M. Impact of macro-level socio-economic factors on rising suicide rates in South Korea: panel-data analysis in East Asia. J Ment Health Policy Econ. 2014;17(4):151-62.

3. Park J, Jeon M. The stigma of mental illness in Korea. Journal of Korean Neuropsychiatric Association. 2016;55(4):299-309.

4. https://www.statista.com/statistics/1230755/south-korea-number-of-suicides-by-reason/

Comments from Reviewer #3: 

1. South Korea has one of the world's highest suicide rates per capita. The reasons for this are not clear, but are likely to be multifactorial. As such, the findings in the authors' manuscript may not be generalizable to a different country with lower per capita suicide rates. I would add this as a limitation.

Answer: Thank you for comments. We fully agree with the reviewer's opinion that our study finding are generally applicable to other countries. Therefore, the following sentence has been added to the limitation part of the discussion:

(Page 16, Line 326- Page 17, Line 331) Finally, our study findings cannot be easily generalized to other countries. South Korea is considered as a developed country with a very rapid economic growth. At the same time, South Korea has the highest suicide rate among OECD countries. While South Korea has the advantage of physically and economically easy access to medical service, there are cultural barriers that are reluctant to visit psychiatry for mental disorder.

---

## [Editor Report · Decision Letter 2]

10 Feb 2022

PONE-D-21-21842R2Association between Benign Prostatic Hyperplasia and Suicide in South Korea: a Nationwide Retrospective Cohort StudyPLOS ONE

Dear Dr. Oh,

Thank you for submitting your manuscript to PLOS ONE. After careful consideration, we feel that it has merit but does not fully meet PLOS ONE’s publication criteria as it currently stands. Therefore, we invite you to submit a revised version of the manuscript that addresses the points raised during the review process.

ACADEMIC EDITOR: Can you pleas add the (new) reference(s), that you provided in your answer regarding the limitations for generalizability, in the text of the manuscript as well?

We look forward to receiving your revised manuscript.

Kind regards,

Peter F.W.M. Rosier, M.D. PhD

Academic Editor

PLOS ONE

Journal Requirements:

Additional Editor Comments:

none
---

## [Author Response · Author response to Decision Letter 2]

21 Feb 2022

Manuscript ID: PONE-D-21-21842.R2

Original title: 

Association between Benign Prostatic Hyperplasia and Suicide in South Korea: a Nationwide Retrospective Cohort Study

To the Editors and Reviewers: 

We thank you for careful reading of the manuscript and helpful comments and suggestions from reviewers. We are submitting reviewers’ comments together, and plan how we revised our paper according to the reviewers’ suggestions and comments. Changes made to the text are marked with highlights changes and track change mode in the revised manuscript.

Comments from Academic Editor: 

1. Can you please add the (new) reference(s), that you provided in your answer regarding the limitations for generalizability, in the text of the manuscript as well? 

Answer: As the editor suggested, we have added new references (no.33- no.34) to the limitation part of external generalization as follows:

(Page 16, Line 326 – Page 17, Line 2) Finally, our study findings cannot be easily generalized to other countries. South Korea is considered as a developed country with a very rapid economic growth [33]. At the same time, South Korea has the highest suicide rate among OECD countries [15]. While South Korea has the advantage of physically and economically easy access to medical service, there are cultural barriers that are reluctant to visit psychiatry for mental disorder [34].

References)

33. Hong J, Knapp M. Impact of macro-level socio-economic factors on rising suicide rates in South Korea: panel-data analysis in East Asia. J Ment Health Policy Econ. 2014;17:151-162.

34. Park JI, Jeon M. The Stigma of Mental Illness in Korea. J Korean Neuropsychiatr Assoc. 2016:55:299-309.

Paper of Hong J et al (No.33) pointed out that suicide rate in South Korea especially higher compared to other Asian countries, despite of similarities in geography and culture. The findings highlight the differential associations between social changes and suicide rates at various stages over a person's life course.

Paper of Park JI (No.34) discussed the stigma of psychiatric disorders in Korea and the reluctance to visit a psychiatric clinic/hospital.

Answer: We checked all references and confirmed that they are correct.

---

## [Editor Report · Decision Letter 3]

23 Feb 2022

Association between Benign Prostatic Hyperplasia and Suicide in South Korea: a Nationwide Retrospective Cohort Study

PONE-D-21-21842R3

Dear Dr. Oh,

We’re pleased to inform you that your manuscript has been judged scientifically suitable for publication and will be formally accepted for publication once it meets all outstanding technical requirements.

Kind regards,

Peter F.W.M. Rosier, M.D. PhD

Academic Editor

PLOS ONE

Additional Editor Comments (optional):

None
---

## [Editor Report · Acceptance letter]

28 Feb 2022

PONE-D-21-21842R3 

Association between Benign Prostatic Hyperplasia and Suicide in South Korea: a Nationwide Retrospective Cohort Study 

Dear Dr. Oh:

I'm pleased to inform you that your manuscript has been deemed suitable for publication in PLOS ONE. Congratulations! Your manuscript is now with our production department. 

Kind regards, 

on behalf of

Dr. Peter F.W.M. Rosier 

Academic Editor

PLOS ONE